# Oropharyngeal Dysphagia and Impaired Motility of the Upper Gastrointestinal Tract—Is There a Clinical Link in Neurocritical Care?

**DOI:** 10.3390/nu13113879

**Published:** 2021-10-29

**Authors:** Paul Muhle, Karen Konert, Sonja Suntrup-Krueger, Inga Claus, Bendix Labeit, Mao Ogawa, Tobias Warnecke, Rainer Wirth, Rainer Dziewas

**Affiliations:** 1Department of Neurology with Institute for Translational Neurology, Albert-Schweitzer-Campus, 1 A, University Hospital Muenster, 48149 Muenster, Germany; karen.konert@uni-muenster.de (K.K.); sonja.suntrup-krueger@ukmuenster.de (S.S.-K.); inga.claus@ukmuenster.de (I.C.); bendixruven.labeit@ukmuenster.de (B.L.); tobias.warnecke@ukmuenster.de (T.W.); 2Institute for Biomagnetism and Biosignalanalysis, University Hospital Muenster, Malmedyweg 15, 48149 Muenster, Germany; 3Department of Rehabilitation Medicine I, School of Medicine, Fujita Health University, Toyoake 470-1192, Japan; positiclub111@yahoo.co.jp; 4Department of Geriatric Medicine, Marien Hospital Herne, University Hospital Ruhr-Universität Bochum, 44625 Herne, Germany; r.wirth@web.de; 5Department of Neurology, Klinikum Osnabrück, Am Finkenhügel 1, 49076 Osnabrück, Germany; rainer.dziewas@klinikum-os.de

**Keywords:** gastric residual volume, dysphagia, flexible endoscopic evaluation of swallowing, gastric emptying, intensive care, neurology, swallowing

## Abstract

Patients in the neurological ICU are at risk of suffering from disorders of the upper gastrointestinal tract. Oropharyngeal dysphagia (OD) can be caused by the underlying neurological disease and/or ICU treatment itself. The latter was also identified as a risk factor for gastrointestinal dysmotility. However, its association with OD and the impact of the neurological condition is unclear. Here, we investigated a possible link between OD and gastric residual volume (GRV) in patients in the neurological ICU. In this retrospective single-center study, patients with an episode of mechanical ventilation (MV) admitted to the neurological ICU due to an acute neurological disease or acute deterioration of a chronic neurological condition from 2011–2017 were included. The patients were submitted to an endoscopic swallowing evaluation within 72 h of the completion of MV. Their GRV was assessed daily. Patients with ≥1 d of GRV ≥500 mL were compared to all the other patients. Regression analysis was performed to identify the predictors of GRV ≥500 mL/d. With respect to GRV, the groups were compared depending on their FEES scores (0–3). A total of 976 patients were included in this study. A total of 35% demonstrated a GRV of ≥500 mL/d at least once. The significant predictors of relevant GRV were age, male gender, infratentorial or hemorrhagic stroke, prolonged MV and poor swallowing function. The patients with the poorest swallowing function presented a GRV of ≥500 mL/d significantly more often than the patients who scored the best. Conclusions: Our findings indicate an association between dysphagia severity and delayed gastric emptying in critically ill neurologic patients. This may partly be due to lesions in the swallowing and gastric network.

## 1. Introduction

The upper gastrointestinal (GI) tract consists of the mouth, pharynx, esophagus, stomach and duodenum. To provide sufficient nutrition and fluid intake, a finely tuned interaction between the structures of the GI tract is crucial [1,2,3,4], starting with the oropharyngeal phase of swallowing. Oropharyngeal dysphagia (OD) is a key feature of different neurological diseases, such as stroke, neuromuscular and neurodegenerative disorders [5]. Particularly in the context of neurocritical care, OD is associated with an increased risk of complications, such as malnutrition and aspiration pneumonia, and is also intimately linked to an overall poor prognosis [2,6,7]. The pathophysiology of OD is complex and may, according to the specific disease in question, involve damage to the central and/or peripheral levels of the swallowing network [2]. Furthermore, in the critically ill, direct trauma to the pharyngeal and laryngeal mucosa caused, for example, by endotracheal or nasogastric tubes, may worsen peripheral sensory feedback and thereby aggravate swallowing impairment [8].

GI motility is also frequently disordered in the critically ill, with up to 60% of patients having been reported to experience GI dysmotility of some form and necessitating therapeutic intervention [3,4]. GI dysmotility of the upper GI tract has significant clinical consequences, being associated with diminished provision of enteral nutrition and subsequent malnutrition, gastroesophageal reflux, and aspiration, as well as longer length of stay (LOS) in the intensive care unit (ICU) and increased mortality [9]. The pathophysiology of GI dysmotility in the critically ill is complex and, to a large extent, still unclear [4]. Interestingly, apart from the consequences of ICU treatment itself and, in particular, the GI side-effects of opioids and sedatives, alterations of hormonal pathways and impaired intrinsic modulation via enteric nerves [10], there is some evidence that dysfunction of the different parts of the nervous system may also contribute to GI dysmotility. Thus, probably because they also cause lesions to the cortical representation of the esophagus [11,12], acute strokes were shown to be related to esophageal dysmotility [13,14] and gastroesophageal reflux [15], ultimately increasing the risk of aspiration and subsequent pneumonia in affected patients [16]. In addition, patients with brain injuries have frequently been reported to present with delayed gastric emptying, resulting in gastric feeding intolerance and its sequelae [17,18,19,20,21,22].

In the present study, therefore, we investigate whether there is a correlation between OD and GI dysmotility, in particular delayed gastric emptying, in a comparatively large cohort of critically ill neurological patients requiring treatment in the ICU and mechanical ventilation (MV).

## 2. Materials and Methods

### 2.1. Study Design and Setting

This retrospective single-center investigation was conducted using the data of patients admitted to the neurological ICU of Münster University Hospital between January 2011 and December 2017. The inclusion criteria were: admittance to the neurological ICU due to an acute neurological disease, or the acute deterioration of a chronic neurological condition, an episode of MV and flexible fiberoptic endoscopic evaluation of swallowing (FEES) within 72 h of the completion of MV (either extubation or, in tracheotomized patients, the completion of weaning). The exclusion criteria were FEES ≥ 72 h after end of MV, palliative care and reduced vigilance (≤8 points on the Glasgow Coma scale), due to its impact on swallowing function. The data were derived from the clinical documentation system.

### 2.2. Patient Characteristics and Clinical Parameters

The epidemiological data, including sex and age, the Body Mass Index, the modified Rankin Scale (mRS) [23] on admission and discharge, the Functional Oral Intake Scale on discharge (FOIS) [24], the RASS (Richmond-Agitation-Sedation-Scale) [25] at the time of initial FEES after the completion of weaning from MV, the Acute Physiology And Chronic Health Evaluation (APACHE) II [26] on admission and discharge, the occurrence of pneumonia [27], sepsis [28] or ileus, the duration of treatment with anti-infectives and, in the case of ischemic or hemorrhagic stroke, the supra- and/or infratentorial lesion location were extracted from the patients’ files. Furthermore, if the volume of enteral nutrition (EN) was reduced and/or prokinetics were administered due to high gastric residual volume (GRV), this was recorded as well.

### 2.3. Dysphagia Assessment

According to our in-house guidelines, all the patients were examined at their bedside in an upright position by an experienced neurologist, together with a speech-language pathologist. The FEES were assessed according to the items ‘secretion management’, ‘spontaneous swallowing’ and ‘laryngeal sensibility/cough^’^. These items were scored, as previously described, according to the “Standardized Endoscopic Swallowing Evaluation for Tracheostomy Decannulation in Critically Ill Neurologic Patients” (SESETD) [29,30]. For this purpose and for better comparability across the patient collective, the items were similarly rated in non-tracheotomized patients as well. The item ‘saliva management’ was considered failed if massive pooling (not only coating) causing an impaired view of the vocal folds and/or silent penetration and/or aspiration of pooled saliva (permanently without any reaction) occurred. ‘Spontaneous swallows’ were considered failed if ≤2 swallows occurred during 2 min of observation. If no reaction to touch of the arytenoids with the tip of the endoscope on both sides could be elicited, the item ‘laryngeal sensibility’ was rated as “not passed”. Deriving from these three single items with passing = 1 point and failing = 0 points, a sum score was built, reaching from 0 to 3, as previously described [30]. All the examinations were part of local routine clinical care. The FEES were carried out using a 3.1-mm-diameter flexible fiberoptic rhinolaryngoscope (11101 RP2, Karl Storz, Tuttlingen, Germany), a combined light source and camera system (rp CAM-X, rpSzene^®^, Rehder/Partner, Hamburg, Germany) and a Medical Panel PC (WMP-226, Wincomm Corporation, Hsinchu, Taiwan) for display and recording. The videos were produced in standard definition quality. The data acquisition and analysis were approved by the local ethics committee.

### 2.4. Evaluation of Gastric Residual Volume

The amount of GRV was recorded daily (6 a.m. to 6 a.m.). For this purpose, the GRV drained into a reservoir connected to the gastric tube following gravity, according to our clinical routine and as previously described [31,32]. The reservoir was connected to the nasogastric tube (NGT) every 12 h for 1 h 30 min after the conclusion of EN. If vomiting or a significant amount of GRV were detected by our nursing staff, the EN was paused for 12 h. The patients were managed in a semi-recumbent position (30–45°) during the drainage of the GRV to prevent aspiration. Patients who had received in vivo thrombolysis and/or thrombectomy or surgery (e.g., external ventricular drainage) were kept nil-by-mouth for the first 24 h and EN was started thereafter. A GRV ≥500 mL/d on at least one day during the stay on the neurological ICU was defined as significant. This cut-off was chosen according to current recommendations and previous studies assuming this amount of GRV to be clinically relevant [31,32].

### 2.5. Statistical Analysis

The characteristics and clinical parameters of patients with vs. without increased GRV were compared. To test for a normal distribution of continuous variables, the Kolmogorov–Smirnov test was applied. For normally distributed data, the t-test was performed for group comparison, otherwise the Mann–Whitney U-test was used. The categorical variables were tested using the Fisher exact test in case the contingency tables included fewer than five cases and the chi^2^-test was used in case of a larger sample size. The significance level was set at 0.05. The significant variables in these univariate analyses were later included in a multivariate binary logistic regression analysis to identify the independent predictors of relevant GRV. The variables that were only gathered at discharge were not included. Pearson correlation was applied to test for an association between initial FEES sum score and days with significant GRV. All the analyses were performed using SPSS 26.0 (IBM, Armonk, NY, USA).

## 3. Results

Of the 1461 patients admitted to the neurological ICU with an episode of MV during the observational period, for further analysis, 295 had to be excluded (see patient recruitment diagram, Figure 1). Hence, 976 patients (423 females) were included in this study (Figure 1) of whom 627 (64.2%) were tracheotomized.

The epidemiological and clinical parameters are summarized in Table 1. On the initial FEES following the conclusion of MV, 360 patients received a score of 0, indicating severe dysphagia (36.9%); 145 passed one of the three items used to evaluate swallowing function (14.9%); and 173 received a score of 2 (17.7%). A total of 297 patients passed all three items (30.4%).

We observed a significant negative correlation between FEES score and the number of days with relevant GRV (Pearson correlation coefficient −0.125, *p* < 0.01). The number of days of GRV ≥500 mL/d according to the initial FEES score after the conclusion of MV can be seen in Figure 2.

Comparing groups without vs. with significant GRV, the latter demonstrated a longer LOS in the ICU (*p* < 0.001) and duration of MV (*p* < 0.001), suffered from hemorrhagic stroke (*p* = 0.004) and infratentorial lesions more often (*p* = 0.014), were younger (*p* < 0.001), included more males (*p* < 0.001), received a lower APACHE II on admission (*p* = 0.008), scored worse on the initial FEES after the conclusion of weaning (*p* < 0.001), including every single item of the sum score, suffered more often from pneumonia (*p* = 0.028) or sepsis (*p* = 0.004) and were discharged from the hospital with PEG or NGT significantly more often (*p* < 0.001). Multivariate logistic regression analysis (Table 2) indicated the following factors as significant independent predictors of GRV ≥500 mL/d on at least one day: dysphagia severity as evaluated by the FEES sum score (*p* = 0.010), duration of MV (*p* = 0.004), hemorrhagic stroke (*p* = 0.042), infratentorial lesion location of stroke (*p* = 0.019), age (*p* < 0.001) and male gender (*p* = 0.018).

## 4. Discussion

In this study, we assessed the relationship between the occurrence and degree of GI dysmotility and OD in ventilated patients in the neurological ICU and tried to identify predictors of GI dysmotility in this cohort. In support of our hypothesis, our first main finding was that the impairment of swallowing function diagnosed at the conclusion of MV was associated with relevant GRV as a surrogate marker of GI dysmotility. While the GI tract possesses intrinsic neural plexus that allow a certain degree of autonomy over digestion and nutrient absorption, the central nervous system provides extrinsic input that regulates, modulates and controls these functions [33]. The small and large intestines exert comparatively independent neural control; the stomach, however, is considerably dependent on extrinsic neural inputs, particularly from the parasympathetic and sympathetic pathways connected to nuclei located in the caudal brainstem [33,34]. In line with this, relevant GRV was observed in patients with an infratentorial lesion significantly more often in our data. Recently, Rebollo et al. identified delayed connectivity between the brain and the slow electrical rhythm generated in the stomach using gastric-BOLD coupling, indicating a functional brain–gut link [35]. Within the brain, different nodes of this ‘gastric network’ were coupled to the gastric rhythm with different phase delays, indicating a temporal sequence of activations within this network—which, in principle, is similar to the central control of the swallowing network [36]. Interestingly, the gastric network partly comprises regions (‘nodes’) that are similarly found to be activated during swallowing, e.g., the primary and secondary somatosensory cortex and the supplementary motor area, as well as the insula [37,38,39]. Thus, the close clinical relation of both functions, in the present study may have been due to lesions that affected swallowing as well as the gastric network.

Our second main finding was that prolonged MV was a significant predictor of impaired gastric emptying, which was in line with previous findings [40]. There are indications that positive pressure mechanical ventilation leads to splanchnic vasoconstriction and gut-hypoperfusion, which is linked with increased plasma catecholamines and proinflammatory cytokine levels, both of which are related to delayed gastric emptying [41]. Furthermore, alterations in hormone levels in the critically ill have an impact on GI motility. Lowered ghrelin levels, as well as increased levels of cholecystokinin and peptide YY, were found in the critically ill and are linked with slower gastric emptying [42,43]. Sedatives such as Propofol and the use of opioids to provide sufficient analgesia during mechanical ventilation were shown to be associated with delayed gastric emptying, as well as the use of catecholamines/vasopressors [3,40,43]. Interestingly, swallowing function has also been shown to be worse in patients with prolonged mechanical ventilation and longer ICU treatment, as well as following the use of sedatives [44], supporting the hypothesis that—at least partly—the underlying mechanisms that cause impaired swallowing function and slowed gastric motility may be similar.

The third main finding was that patients suffering from intracranial hemorrhage seem to be at a particularly high risk of slowed gastric emptying. It was previously shown that intracranial hemorrhage is a risk factor for inferior swallowing function compared to ischemic stroke in patients with and without tracheostomy [45,46]. It was proposed that besides the specific localization [47] and volume of the intracranial hemorrhage [48], this may at least partly be attributed to secondary consequences of the hemorrhage, e.g., vasospasms [49], cisternal and interventricular blood or hydrocephalus [48]. There is more evidence that increased intracranial pressure is related to delayed gastric emptying. Thus, in a study of 21 brain-injured patients requiring sedation, MV and intracranial pressure monitoring for ≥24 h, increased intracranial pressure (>20 mmHg) was associated with reduced gastric emptying, as measured by the paracetamol absorption technique, possibly due to a decreased parasympathetic tonus [18]. In a study by Kao et al., 80% of head-injury patients exhibited abnormal gastric emptying halfway through liquid meals compared to healthy age-matched control subjects [20]. Using electrogastrography, it was further shown that brain trauma or coma cause gastric dysrhythmias and intolerance to feeding, supporting the hypothesis of an altered functional brain–gut link that causes delayed gastric emptying in patients with acquired brain injury [21].

As our fourth and fifth main findings, younger age and male gender were demonstrated to be related to delayed gastric emptying. Findings on the effects of ageing in the context of gastric motor function are inconsistent. Studies in healthy as well as critically ill patients found indications of declining gastric motor function with increasing age [43,50], whereas others identified a trend towards increased gastric emptying depending on increasing age in healthy adults [51]. In the neuro-ICU setting, as mentioned above, the influence of the intensive care treatment as well as the underlying condition causing the need for treatment may have caused the differing findings in the recent study. This can similarly be assumed for the gender differences. In general, gastric motility seems to be slower in healthy women than in healthy men [52] but there are indications that gastric motility may be less gender-specific depending on the consistency of administered boluses [53]. For a better understanding, the role of age as well as gender in the intensive care setting needs be investigated more closely.

The clinical relevance of GRV is still a matter of discussion. Generally, the intermittent measurement of GRV is a widely used practice to evaluate impaired gastric motility and feeding intolerance [4]. Several studies, including a meta-analysis, indicated that not monitoring GRV was not inferior to routine GRV measurement with regard to ICU-related infections, LOS in the ICU, length of MV and mortality. Furthermore, not monitoring GRV even improved the delivery of enteral nutrition [54,55,56]. These data were mainly derived from mixed cohorts. In stroke patients, who often suffer from dysphagia and impaired protective reflexes, Chen et al. observed that aspiration occurred significantly less often if GRV was monitored and the infusion rate of the EN was adjusted accordingly [57]. In our cohort of critically ill neurologic patients, relevant amounts of GRV were associated with pneumonia and sepsis. Pneumonia in the context of delayed gastric emptying in the critically ill is thought to be caused by aspiration due to gastro-esophageal reflux, which itself is a consequence of reduced esophageal sphincter tonus and increased residual volume in the stomach [4]. Dysphagia is another risk factor for pneumonia, notably as a result of aspiration [2]. Since patients with relevant GRV presented with a worse swallowing function, both disorders may foster each other. In line with this, patients with GI dysmotility presented a worse FOIS score at discharge and were more likely to be discharged with a feeding tube. Moreover, during systemic inflammation, intestinal edema deriving from capillary leakage influences GI function and cytokine release during sepsis, impedes intestinal myocyte function and inhibits enteric neuromuscular transmission [58,59,60,61,62].

Certain limitations to our study should be considered. First, the retrospective design may have introduced a bias into our data, which possibly include imprecise documentation of the patients’ records. Second, all the patients were recruited on a single neurological ICU; hence, the transfer of findings to other environments and, in particular, to groups of patients with a different spectrum of diseases may be only be possible only to a limited extent. Third, bedside measures were previously shown to be imprecise in the identification of motility disorders [63]. While the intermittent measurement of GRV may be the most common practice through which gastric motility disorders are evaluated, indirect tests, such as the carbohydrate absorption (3-OMG), the radio-isotope breath (_13_CO_2_) or the aforementioned paracetamol absorption test, as well as gastric scintigraphy, evaluate gastric dysmotility with more precision, although they are not always applicable in the ICU setting [4]. Fourth, with regards to the impact of our findings, no long-term outcome assessment was available.

## 5. Conclusions

The findings in this study indicate an association between delayed gastric emptying and dysphagia severity in critically ill neurologic patients in the ICU. Beside the effects of intensive care treatment, there are indications that central lesions in the swallowing and gastric network both add to the deterioration of swallowing function as well as to the impairment of upper GI motility.

## Figures and Tables

**Figure 1 nutrients-13-03879-f001:**
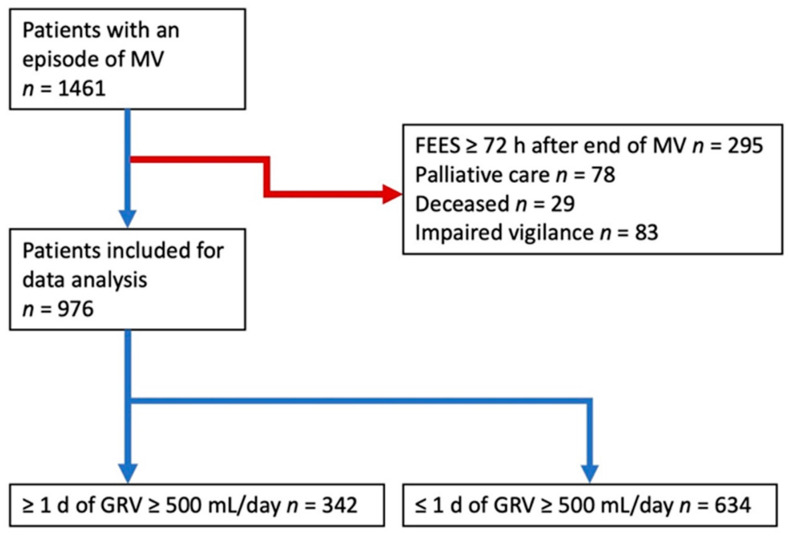
Recruitment flowchart; MV = mechanical ventilation; FEES = flexible endoscopic evaluation of swallowing; GRV = gastric residual volume; mL = milliliters.

**Figure 2 nutrients-13-03879-f002:**
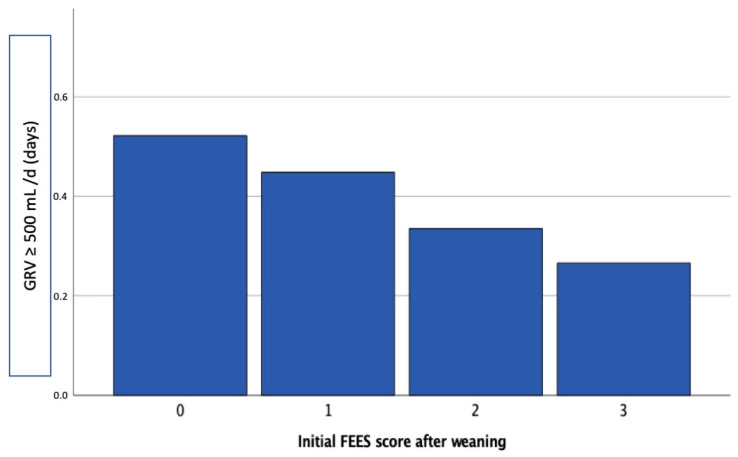
Days of gastric residual volume ≥500 mL/d, according to sum score on the first flexible endoscopic evaluation of swallowing (FEES) within 72 h of the conclusion of mechanical ventilation. Score 0: *n* = 360; Score 1: *n* = 146; Score 2: *n* = 173; Score 3: *n* = 297; GRV = gastric residual volume; mL = milliliters.

**Table 1 nutrients-13-03879-t001:** Epidemiological and clinical parameters and group test according to GRV.

	All *n* = 976	Max. GRV < 500 mL/d *n* = 634 (65.0%)	Max. GRV ≥ 500 mL/d *n* = 342 (35.0%)	*p*-Value
**Age, mean (SD)**	64.79 (±16.06)	66.78 (±16.06)	61.08 (±15.41)	<0.001 †
**Female/Male, *n* (%)**	423 (43.3)/553 (56.7)	301 (47.5)/333 (52.5)	122 (35.7)/220 (64.3)	<0.001 ‡
**Body mass index, mean (SD)**	26.61 (±5.15)	26.41 (±4.88)	26.92 (±5.59)	0.116 †
**Ischemic stroke, *n* (%)**	546 (55.9)	355 (60.0)	191 (55.8)	0.932 ‡
**Hemorrhagic stroke, *n* (%)**	155 (15.9)	85 (13.4)	70 (20.5)	0.004 ‡
**Lesion location strokes**				
**Supratentorial, *n* (%)**	569 (58.3)	367 (57.9)	202 (59.1)	0.722 ‡
**Infratentorial, *n* (%)**	132 (13.5)	73 (11.5)	59 (17.3)	0.014 ‡
**Meningitis/Encephalitis, *n* (%)**	76 (7.8)	55 (8.7)	21 (6.1)	0.159 ‡
**GBS/AMAN, *n* (%)**	24 (2.5)	15 (2.4)	9 (2.6)	0.798 ‡
**Myopathy/Myasthenia/Myositis, *n* (%)**	13 (1.3)	9 (1.4)	4 (1.2)	1.000 §
**Epilepsy, *n* (%)**	82 (8.4)	58 (9.1)	24 (7.0)	0.252 ‡
**Amyotrophic lateral sclerosis, *n* (%)**	11 (1.1)	9 (1.4)	2 (0.6)	0.346 §
**Others, *n* (%)**	69 (7.1)	46 (7.3)	23 (6.7)	0.758 ‡
**mRS on admission, mean [median]**	4.57 [5 (4–5)]	4.59 [5 (4–5)]	4.54 [5 (4–5)]	0.135 †
**APACHE II on admission, mean [median]**	13.67 [13 (10–17)]	14.02 [14 (10–18)]	13.04 [13 (9–17)]	0.008 †
**Mechanical ventilation (h), mean (SD)**	334.05 (±355.18)	264.88(±314.32)	462.28 (±389.77)	<0.001 †
**LOS ICU (d), mean (SD)**	27.94 (±20.62)	23.71 (±19.33)	35.76 (±20.67)	<0.001 †
**FEES sum score after end of MV, mean [median]**	1.42 [1 (0–3)]	1.55 [2 (0–3)]	1.17 [1 (0–2)]	<0.001 †
**Aspiration/pooling, *n* (%)**	463 (47.4)	269 (42.4)	194 (56.7)	<0.001 ‡
**Swallowing frequency <2x/2 min, *n* (%)**	457 (46.8)	268 (42.3)	189 (55.3)	<0.001 ‡
**Failing sensory testing, *n* (%)**	625 (64.0)	381 (60.1)	244 (71.3)	<0.001 ‡
**Antiinfective treatment (d), mean (SD)**	19.71 (±13.73)	17.40 (±13.34)	23.99 (±14.26)	<0.001 †
**Pneumonia, *n* (%)**	691 (70.8)	434 (68.5)	257 (75.1)	0.028 ‡
**Sepsis, *n* (%)**	78 (8.0)	39 (6.2)	39 (11.41)	0.004 ‡
**Diabetes mellitus, *n* (%)**	226 (23.2)	154 (24.3)	72 (21.1)	0.253 ‡
**Medication due to high GRV, *n* (%)**	465 (47.8)	203 (32.0)	334 (97.7)	<0.001 ‡
**NGT/PEG on discharge, *n* (%)**	533 (58.7)	318 (54.1)	215 (67.2)	<0.001 ‡
**FOIS at discharge, mean [median]**	3.25 [3 (1–5)]	3.49 [3 (1–6)]	2.83 [2 (1–5)]	<0.001 †
**Deceased on ICU, *n* (%)**	59 (6.0)	38 (6.0)	21 (6.1)	0.927 ‡
**mRS at discharge, mean [median]**	4.33 [5 (4–5)]	4.29 [5 (4–5)]	4.41 [5 (4–5)]	0.168 †

SD = standard deviation; h = hours; d = days; LOS = length of stay; ICU = intensive care unit; GRV = gastric residual volume; GBS = Guillain-Barré syndrome; AMAN = acute motor axonal neuropathy; mRS = modified Rankin Scale; FEES = flexible endoscopic evaluation of swallowing; EN = enteral nutrition; NGT = nasogastric tube, FOIS = Functional Oral Intake Scale; † = Mann–Whitney U-test; ‡ = chi^2^-test; § = Fisher-exact test.

**Table 2 nutrients-13-03879-t002:** Multivariate binary logistic regression analysis; outcome variable: GRV ≥ 500 mL/d on at least one day.

	Regression Coefficient	Adjusted Odds Ratio [95% CI]	*p*-Value
**Age**	−0.019	0.981 [0.971–0.991]	**<0.001**
**Male gender (cat.)**	0.351	1.421 [1.061–1.903]	**0.018**
**Hemorrhagic stroke (cat.)**	0.394	1.483 [1.015–2.166]	**0.042**
**Infratentorial lesion location (stroke) (cat.)**	0.491	1.634 [1.083–2.465]	**0.019**
**Mechanical ventilation (hours)**	0.001	1.001 [1.001–1.002]	**0.004**
**LOS on the ICU (days)**	0.012	1.102 [0.996–1.028]	0.146
**FEES sum score initial FEES after end of weaning (cat.)**	−0.155	0.857 [0.762–0.963]	**0.010**
**APACHE II**	−0.025	0.975 [0.948–1.002]	0.073
**Days of antiinfective treatment**	0.004	0.996 [0.978–1.015]	0.688
**Sepsis (cat.)**	0.322	1.380 [0.820–2.324]	0.226

cat = categorical; LOS = length of stay; ICU = Intensive Care Unit.

## Data Availability

The data presented in this study are available on request from the corresponding author. The data are not publicly available due to hospital policy.

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
