# Peer review of "Oropharyngeal Dysphagia and Impaired Motility of the Upper Gastrointestinal Tract—Is There a Clinical Link in Neurocritical Care?"

_nutrients, 2021, doi:10.3390/nu13113879_

Round 1
Reviewer 1 Report
This paper is well written across the board. In my opinion, authors can understand correlation of OD and GI dysmobility according to the background, methods, results, discussion, and conclusions you present. But, I have some questions and minor comments related to statistical analysis for better contributions and more accuracy.
- In Figure 1, you finally selected 976 patients. I wonder there are no outliers in these patients for some reasons (e.g., missing data, considerable big/small values at corresponding factor).
- In Table 1, only Mann-Whiteny U-test results can be found for group comparisons. But, you wrote t-test was performed for normally distributed data. It means there is no normally distributed data. Can you check again because your sample size was large (N = 976) so that possibility of normal distribution is high?
- In Table 2, you found significant factors by applying multivariate binary logistic regression analysis. In terms of categorical variables, can you present differences among factor levels (averages or figures)? Those information can be helpful for better understanding of significant predictors of GRV≥500 ml/d.
Author Response
Thank you very much for reviewing this article and your positive evaluation. Also thank you for your valuable suggestions to further improve its content and structure:
- In Figure 1, you finally selected 976 patients. I wonder there are no outliers in these patients for some reasons (e.g., missing data, considerable big/small values at corresponding factor).
Ad 1) Thank you for pointing out this relevant aspect. We used a digital clinical documentation system, called QS which nurses, SLTs and physicians use for daily documentation. Here, epidemiological data as well as all clinical data can be entered, to a certain amount even automatically with data being derived directly from the patient monitor (e.g. heart rate) etc. The data that were used for this study are mainly part of our daily documentation, e.g. GRV and medication and are therefore checked at least three times daily during rounds. If data from this system were not sufficient, we could also use data from our second clinical documentations system including data from procedures and treatment outside of the ICU. Furthermore, medical reports from general practitioners as well as other hospitals is scanned and can be searched using this system. We therefore expect that a relatively high quality of documentation has been achieved but – due to this being a retrospective study – we cannot guarantee that some of the documentation was imprecise. We added the following sentence to our limitations section:
First, the retrospective design may have introduced a bias into our data which possibly includes imprecise documentation in patients’ records.
- In Table 1, only Mann-Whitney U-test results can be found for group comparisons. But, you wrote t-test was performed for normally distributed data. It means there is no normally distributed data. Can you check again because your sample size was large (N = 976) so that possibility of normal distribution is high?
Ad 2) This indeed seemed odd to us as well. We rechecked our data but despite the relatively large number of participants normal distribution was not given in the corresponding variables in table 1 according to the Kolmogorov-Smirnov test.
- In Table 2, you found significant factors by applying multivariate binary logistic regression analysis. In terms of categorical variables, can you present differences among factor levels (averages or figures)? Those information can be helpful for better understanding of significant predictors of GRV≥500 ml/d.
Ad 3) Thank you for pointing this out. All categorical variables used in the regression analysis our now presented in detail in figure 3 (supplement). If the reviewer and/or editor wishes this figure to be included in the article we can add it, e.g. as supplement.

Reviewer 2 Report
The manuscript by Paul Muhle et al described the link between oropharyngeal dysphagia and gastric residual volume in patients in the neurological ICU. The manuscript is well written and descriptive, however, it has several concerns.
Result and discussion
- Line 154, what are the meaning of one item and all three items? The word item does not seem clear.
- How to determine the parameters on table 1 and 2?
- Is there any specific reason why the author included diabetes mellitus as the parameter in table 1?
- In materials and methods, the experimentation data was collected from 2011 to 2017. Why the author did not use the recent data, for example 2013 to 2017?
- It would be interesting for the future study if the author adds the genetic family history/disease as a parameter. Since, there are some articles stated that OD has something to do with a genetic.
Author Response
Reviewer 2
Thank you very much for reviewing our manuscript and your valuable suggestions to improve the results and discussion section of this article.
Result and discussion
- Line 154, what are the meaning of one item and all three items? The word item does not seem clear.
Ad 1) In this context, we use the term item to describe the three aspects of the score used to evaluate swallowing function: “secretion management”, “spontaneous swallowing” and “laryngeal sensibility/cough”. This is described in “dysphagia assessment” in the methods part of the manuscript. However, it is correct that in the results section this appears a little out of context. To make it more easily understandable, we changed line 154 to:
On the initial FEES following the end of MV, 360 patients had a score of 0 indicating severe dysphagia (36.9 %), 145 passed one of the three items used to evaluate swallowing function (14.9 %) and 173 had a score of 2 (17.7 %). 297 patients passed all three items (30.4 %)
- How to determine the parameters on table 1 and 2?
Ad 2) We used a digital clinical documentation system, called QS which nurses, SLTs and physicians use for daily documentation. Here, epidemiological data as well as all clinical data can be entered, to a certain amount even automatically with data being derived directly from the patient monitor (e.g. heart rate) etc. Table 1 was derived directly from this system by hand and data were grouped according to the occurrence of GRV ≥ 500 ml on at least one day. We then performed group comparison, using the Mann-Whitney U-test for non-normally distributed data and the Chi-square test for categorical variables. Deriving from these findings, all variables that showed to be significant at a level of p<0.05 were included in a binary logistic regression analysis to identify predictors of relevant GRV. The findings from the regression analysis are reported in table 2.
- Is there any specific reason why the author included diabetes mellitus as the parameter in table 1?
Ad 3) In the context of increased gastric residual volume, there are indications diabetes is related to a delayed gastric emptying (e.g. Sabry et al. Evaluation of gastric residual volume in fasting diabetic patients using gastric ultrasound. 2019. Anaesthesiologica Scandinavica). We therefore wanted to present these findings from our relatively large patient collective as well.
- In materials and methods, the experimentation data was collected from 2011 to 2017. Why the author did not use the recent data, for example 2013 to 2017?
Ad 4) We tried to collect as many data as possible up to the end of 2017 when our doctoral candidate started to work on this project. Our current clinical documentation system ‘QS’ was launched in our department in 2011, hence explaining the beginning.
- It would be interesting for the future study if the author adds the genetic family history/disease as a parameter. Since, there are some articles stated that OD has something to do with a genetic.
Ad 5) This is an indeed very interesting aspect that needs further exploration. With recent findings on the influence of genetics on swallowing function (e.g. Raginis-Zborowska, Pendleton and Hamdy. Genetic determinants of swallowing impairment, recovery and responsiveness to treatment. 2016. Curr Phys Med Rehabil Rep) this should be investigated more closely in the context of a possible link between swallowing function and increased GRV. Considering our next steps in the investigation of the link between GRV and swallowing function, we will try to include genetics.
